# Generation of Novel Tumour-Selective SEA Superantigen-Based Peptides with Improved Safety and Efficacy for Precision Cancer Immunotherapy

**DOI:** 10.3390/ijms25179423

**Published:** 2024-08-30

**Authors:** Sara S. Bashraheel, Haya Al-Sulaiti, Sayed K. Goda

**Affiliations:** 1College of Medicine, QU Health, Qatar University (QU), Doha P.O. Box 2713, Qatar; sara.bashraheel@qu.edu.qa; 2College of Health and Science, QU Health, Qatar University (QU), Doha P.O. Box 2713, Qatar; haya.alsulaiti@qu.edu.qa; 3Biomedical Research Center, Qatar University, Doha P.O. Box 2713, Qatar; 4College of Science and Technology, University of Derby, Derby DE22 1GB, UK

**Keywords:** tumour-targeted superantigen, SEA superantigen, cancer combination therapy, targeted cancer immunotherapy, superantigen peptide agonists, superantigen peptide agonist conjugates

## Abstract

Bacterial superantigens are T-cell-stimulatory protein molecules which produce massive cytokines and cause human diseases. Due to their ability to activate up to 20% of resting T-cells, they have effectively killed T-cell-dependent tumours in vivo. However, the intrinsic toxicity of whole SAg molecules highlights the urgent need to develop more effective and safer SAg-based immunotherapy. With its unique approach, our study is a significant step towards developing safer tumour-targeted superantigen peptides (TTSP). We identified the T-cell activation function regions on the SEA superantigen and produced variants with minimal lethality, ensuring a safer approach to cancer treatment. This involved the creation of twenty 50-amino-acid-long overlapping peptides covering the full-length SEA superantigen (P1-P20). We then screened these peptides for T-cell activation, successfully isolating two peptides (P5 and P15) with significant T-cell activation. These selected peptides were used to design and synthesise tumour-targeted superantigen peptides, which were linked to a cancer-specific third loop (L3) of transforming growth factor-α (TGF-α), TGFαL3 from either a C’ or N’ terminal with an eight-amino-acid flexible linker in between. We also produced several P15 variants by changing single amino acids or by amino acid deletions. The novel molecules were then investigated for cytokine production and tumour-targeted killing. The findings from our previous study and the current work open up new avenues for peptide-based immunotherapy, particularly when combined with other immunotherapy techniques, thereby ensuring effective and safer cancer treatment.

## 1. Introduction

Cancer is the second leading cause of death worldwide after cardiovascular disease. highlighting the necessity for innovative therapeutic approaches. Conventional treatments such as chemotherapy, radiation, and surgery are associated with significant side effects due to their toxicity and un-specificity of killing healthy as well as tumour cells [1]. The advances in targeted cancer therapy and personalised medicine, where treatment approaches are tailored to each patient’s unique genetic and molecular characteristics, can overcome many side effects of conventional treatments.

One of the targeted treatments is cancer immunotherapy, which has emerged as a groundbreaking approach that boosts patients’ immune systems to recognise and fight cancer. Recent immunotherapies, including checkpoint inhibitors, CAR T-cell therapy, and cancer vaccines, have demonstrated significant success in treating various types of cancer [2,3,4,5,6,7].

Many patients with solid tumours do not benefit from modern immuno-oncology treatments such as vaccines, immune checkpoint blockades, and many others. This can be explained by limited tumour T-cell activation resulting from low immunogenicity and immune suppression [8,9].

One class of molecules which can play a pivotal role in targeted cancer immunotherapy is superantigens. Superantigens (SAgs) are microbial toxins that promote unspecific T-cell activation by crosslinking between MHC-II on antigen-presenting cells (APC) and the Vβ domain of the T-cell receptor. This sort of binding leads to the activation of up to 20% of the T-cell population and massive cytokine secretion. Thus, SAgs have shown promising results in T-cell-dependent cancer immunotherapy [10,11,12].

However, SAgs are also associated with severe side effects, such as hypotension and symptoms of septic shock syndrome. Our previous work used a different approach to produce novel tumour-targeted SAg-based peptides for tolerable cancer immunotherapy. In our earlier studies, we produced peptide agonists from the full superantigen SPEA capable of T-cell-dependent killing of cancer. We also identified three SPEA-based peptides with a vasodilatory effect, which allowed us to design a superantigen without the hypotension effect on the patient [13,14].

This work employs different superantigens to increase the varieties of superantigen-based peptides for cancer immunotherapy. Staphylococcal enterotoxin type A (SEA) is an effective tumour immunotherapeutic protein. It was shown that when SEA is conjugated with a single-chain Fv (SEA–scFv), an important reagent for targeted cancer immunotherapy, it enhances the cytotoxicity of lymphokine-activated killer cells with a T-cell phenotype against a human bile duct carcinoma cell line. This finding using the SEA–scFv fusion protein could make the SEA superantigen a useful reagent for cancer immunotherapy [15].

In another study, SEA, when co-cultured with human peripheral blood mononuclear cells (PBMCs), inhibited the proliferation and induced the killing of human lung carcinoma A549 cells [16].

It was reported that the transmembrane-SEA superantigen (TM-SEA) cellular vaccine could cause tumour-specific immunity, which suggests that an SEA-based vaccine is an effective strategy for cancer immunotherapy [17].

All the above treatments and many others were achieved with severe toxicity using the whole SEA superantigen. Therefore, a novel form of superantigen-based molecules with tolerable toxicity is needed. In this work, we propose to produce SEA-based peptide agonists with superantigenicity-positive and toxicity-tolerable properties for cancer immunotherapy.

The binding of superantigens with the MHC class II and T-cell receptor results in the activation of the T-cells, which leads to a massive release of cytokines such as IL-2, IFN-γ, TNF-α, and others, generating strong T-cell cytotoxic capacity, which could be employed for the killing of the MHC II-positive tumour cells [18]. However, not all tumour cells express MHC class II, which renders SEA ineffective against many types of cancer. To overcome this problem, a conjugate of wild-type SEA and tumour antigen-specific antibodies was constructed [19,20,21]. The approach was successful; however, the high affinity of SEA for MHC class II has led to the retention of Ab–SEA conjugate proteins in normal tissue expressing MHC II, which causes systemic immune activation and severe side effects [22]. Therefore, to lower systemic immune activation, we produced SEA-based peptide agonists with significantly decreased affinity to MHC II and demonstrated their cancer-killing capabilities.

We designed and synthesised a series of 20 overlapping peptides spanning the entire SEA amino acid sequence for T-cell activation and production of different cytokines:

P1(1–50), P2(11–60), P3(21–70), P4(31–80), 5(41–90), P6(51–100), P7(61–110), P8(71–120), P9(81–130), P10(91–140), P11(101–150), P12(111–160), P13(121–170), P14(131–180), P15(141–190), P16(151–200), P17(161–210), P18(171–220), P19(181–230), and P20(191–242) (Table 1).

The main feature of a superantigen is the nonspecific activation of T-cells, which results in polyclonal T-cell activation and massive release of cytokines such as interferon-gamma, IL-1, IL-6, IL-10, and TNF-alpha.

We conducted our research using a systematic approach. We started by testing all the above peptides for T-cell activation and production of cytokines in a methodical attempt to identify peptides with superantigenicity character. This systematic approach ensures the reliability and validity of our findings.

Human epidermal growth factor receptor (EGFR) plays a pivotal role in cell growth and is highly expressed in cancer cells [23]. It was shown that EGFR is overexpressed in non-lung cancers, colorectal cancer, and breast cancer [24,25,26].

Many studies have initiated targeted cancer therapy strategies based on the specific overexpression of EGFR to cancer cells to concentrate the drug’s toxic effect in the vicinity of the tumour. This was achieved by conjugating different drug molecules, such as antibodies [27,28,29,30] or full superantigen SEB [31], with the third loop (L3) of transforming growth factor-α (TGF-α).

In this work, we achieve specificity with lower-toxicity tumour-targeted superantigen treatment (TTS). Our newly isolated SEA-based peptide agonist will be conjugated with TGFαL3. To avoid any misfolding effect of the loop on the peptide, we synthesised two forms of each peptide by adding the loop on either the N-terminal or C-terminal with the addition of a flexible linker.

The selected peptides were conjugated with cancer-targeted sequence loop TGFαL3. Therefore, the novel tumour-targeted SAg-based peptide conjugates produced could accomplish tolerable cancer immunotherapy.

## 2. Results

### 2.1. Assessment of T-Cell Activation

SEA-based peptides were evaluated for their ability to stimulate T-cells. Treated PBMCs were stained with fluorescent anti-CD25 and anti-CD3 antibodies and screened using a flow cytometer to measure CD3+CD25+ cells. SEA was a positive control, while DMSO and H_2_O were the negative controls. Figure 1A shows that only peptides P5 and P15 significantly affected T-cells. Up to 15% of PBMCs were activated T-cells when treated with P5, whereas P15 resulted in 30% of activated T-cells. Figure 1B shows dot plots of CD3+ and CD25+ gating and selected peptides with positive and negative controls.

Based on the results of Figure 1, we produced eight derivatives of the P15 peptide. T-cell activation assessment was also carried out using these peptides. Figure 2A shows a significantly lower percentage of activated T-cells in response to P15-4 and P15-8 compared to P15. The remaining peptides P15-6, P15-10, P15-D17A, P15-D34A, P15-D37A, and P15-D3A3 had no significant difference in T-cell activity compared to P15 peptide (Figure 2A,B).

### 2.2. Tumour Cell Binding Assay

Selected peptides linked to TGFαL3 legend, constructed as shown in Figure 3, were screened for binding to EGFR-expressing MDA-MB-468 cells using ArrayScan™ XTI instrument from Thermo Fisher Scientific (Waltham, MA, USA). P5TGF, TGFP14, and P14TGF showed the highest percentage of binding to tumour cells, with around 30%, 31%, and 35% binding, respectively. Meanwhile, TGFP15 and P15TGF had slightly lower yet significant binding effects, with around 25% and 20% binding, respectively. Moreover, TGFP5 did not bind significantly to tumour cells (Figure 4).

### 2.3. Antitumour Activity

The viability of tumour cells was determined by counting the DAPI-positive nuclei in each sample and calculating this as a percentage of untreated samples. Tumour killing was detected in response to P5, P15, TGFP5, P5TGF, TGFP14, TGFP15, and P15TGF (Figure 5). P5 showed 60% tumour killing while its two conjugates, TGFP5 and P5TGF, had a slightly reduced antitumour activity of 45% and 35% killing effects, respectively. P14 and its conjugate P14TGF did not have a tumour-killing effect; however, TGFP14 showed a 30% tumour-killing effect. Finally, P15 resulted in up to 50% tumour killing, whereas its conjugates TGFP15 and P15TGF resulted in 40% tumour killing.

Tumour cell viability in response to peptide P15 derivatives was also measured. Figure 6 shows that treatment with P15-6, P15-8, and P15-10 resulted in increased cell viability compared to P15. Peptides P15-4, P15-D17A, P15-D34A, P15-D37A, and P15-D3A3 maintained the same level of tumour cell viability as P15.

### 2.4. Detection of Cytokines

Compared to the solvent control, P5 produced significant amounts of IL-1β, IL-6, and IL-10 but did not elicit TNFα release. In contrast, P5 conjugates TGFP5 and P5TGF produced IL-1β, IL-6, IL-10, and TNFα. P14 and its conjugates TGFP14 and P14TGF stimulated the production of significant amounts of IL-1β, IL-6, IL-10, and TNFα. Similarly, P15 produced IL-1β, IL-6, IL-10, and TNFα significantly. However, the P15 conjugates TGFP15 and P15TGF both produced IL-1β, IL-6, and TNFα, but only P15TGF produced IL-10 (Figure 7).

Figure 8 indicates that all P15 derivatives produced significant amounts of cytokines IL1β, IL-6, and TNFα but with variation in the level compared to P15. The level of IL-1 was elevated in response to P15-D34A, P15-D37A, and P15D3A3 compared to P15, while it decreased in response to P15-6, P15-8, and P15-10. Furthermore, the IL-6 level increased in samples treated with P15-D34A and P15-D37A, whereas it was reduced in samples treated with P15-6, P15-8, and P15-10. Interestingly, the level of TNFα was decreased in all P15 derivatives, and similarly, IL-10 was decreased in all P15 derivatives except P15-D34A.

### 2.5. MTT Assay

The cytotoxicity of the conjugated peptides on PBMCs was evaluated using MTT assays and compared to their solvent controls and the full superantigen. Among all peptide conjugates, TGFP5 exhibited no cytotoxic effect on PBMCs. Figure 9 shows that P14 and P15 peptide conjugates as well as P15TGF caused a significant increase in the absorbance compared to the solvent control, which indicates cell proliferation. However, when compared to the full superantigen, these peptide conjugates caused a reduced negative cell proliferation effect; therefore, they were less toxic to PBMCs.

## 3. Discussion

Due to tumours’ genetic and phenotypical heterogeneity, one strategy may not effectively treat them. Combining different strategies, including traditional anticancer drugs such as chemotherapy and radiotherapy, would lead to more effective treatment and more prolonged overall cancer survival than single-drug therapy.

This work aims to add one more tool to the cancer therapy arsenal: novel superantigenicity-positive lethality-negative SEA-based peptide agonists and peptide agonists conjugated with cancer-specific legend for cancer immunotherapy.

Figure 1 and Figure 7 show that P5 and P15, with the amino acid sequence shown in Table 1, could activate and proliferate human T-cells and produce several cytokines, IL-1, IL-6, IL-10, and TNF-α.

We have demonstrated that P5 and P15 behave as noted by the full SEA superantigen in activating the T-cell and producing different kinds of cytokines. However, they produce a smaller quantity of cytokines than the full SEA, making these peptides safer to use for cancer immunotherapy than the full molecule.

It was shown that the CD28 homodimer interface is a critical receptor target for superantigens through the binding site TNKKMVTAQELD [32]. This amino acid sequence is crucial for the production of cytokines. Table 1 shows that P15 accommodates this essential amino acid sequence. However, this amino acid sequence is not included in the P5, which might indicate another binding site of the superantigen to the T-cells. Our data have shown that two SEA-based peptide agonists, P5 and P15, have a similar profile of the full superantigens and in managing to activate the T-cells, which leads to the release of IL-1, IL6, and IL-10, which can inhibit the proliferation of different kinds of cancer cells. Regarding IL-1, a study has shown that IL-1 inhibits cell proliferation of A375 melanoma, prostate stem cells, and MCF-7 breast cancer murine primary mammary cells by causing G0–G1 cell cycle arrest [33,34,35]. IL-10 also plays a pivotal role in the killing of different cancers [36].

We embarked on a peptide optimisation study using two strategies, size optimisation and amino acid mutation, to obtain a better peptide for T-cell activation and for future work on peptide bioavailability as a drug.

We synthesised different peptide lengths of P15 as P15-4, P15-6, P15-8, and P15-10 (-4, -6, -8, and -10 indicate the removal of 4, 6, 8, or 10 amino acids from the N terminal, respectively). The data of T-cell activation in Figure 2A show a significantly lower percentage of activated T-cells in response to P15-4 and P15-8 compared to P15. For the remaining peptides, P15-6 and P15-10, we could not detect a considerable difference in T-cell activity compared to the P15 peptide. Also, all the amino acid mutations we carried out as shown in Table 2, P15-D17A, P15-D34A, P15-D37A, and P15-D3A3, had no significant difference in T-cell activity compared to the P15 peptide (Figure 2B). Further investigation is needed for the P15 optimisation.

Despite the advances in immunotherapies such as immune checkpoint blockade (ICB), many types of tumours do not benefit from these strategies. Many solid tumours lack tumour-specific T-cell activation and T-cell infiltration [37]. Studies have shown that using tumour-targeted superantigen treatment (TTS) using SEA conjugate, C215Fab-SEA, in isolation or combined with PD-1/PD-LI, led to a long-term antitumour immune response [38] with severe side effects due to the use of the full SEA superantigen.

The data in Figure 2 show that P15-4 has less T-cell activation than P15; however, P15-4 shows a lower cell count than P15 in Figure 6A. This unexpected difference could be explained by the fact that not only the peptide’s structure and sequence but also its binding and folding determine its activity. Also, superantigens or superantigen agonists, after binding to T-cells, induce the production of a broad range of cytokines, including tumour necrosis factor (TNF), interleukin 2 (IL-2), interleukin 1 (IL-1), IL-4, IL-6, and gamma interferon. Although P15-4 produces less T-cell activation, it might stimulate the production of cytokines in a slightly different ratio than P15, which could be an avenue for more effective cancer cell killing.

We successfully report that these peptides and peptide conjugates, P5, P15, TGFP5, P5TGF, TGFP14, TGFP15, and P15TGF, can kill cancer with a much-reduced toxicity effect on PBMCs (Figure 5), as shown in the results section.

We have reported isolating and identifying SEA-based peptides with superantigenicity activity for the first time. When these peptides are conjugated with a cancer-specific ligand, they are capable of killing cancer with no effect on control cells.

The SPEA and SEA superantigens share low amino acid homology. The peptides produced in our previous work on SPEA and the current study on SEA with T-cell activation and the capability of killing cancer cells could be used consecutively in cancer treatment to minimise immunogenicity and drug resistance. In addition, we carried out new work in the current manuscript, such as peptide modifications and peptide mutagenesis for peptide optimisation.

Combination cancer therapy involves two or more therapeutic drugs and is considered the backbone of cancer therapy [39,40]. Combination therapy including the traditional therapies such as chemotherapy, radiotherapy, and surgery boosts the efficacy and efficiency of the treatment because each approach targets different key pathways in a synergistic or additive manner.

Figure 10 shows many of the known strategies for treating different types of cancer. Our novel superantigen peptide agonists and their conjugates from this work and our previous work [14] add one more robust strategy: tumour-targeted superantigen peptides (TTSP) for cancer immunotherapy. We propose that a combination cancer therapy could be carried out using our novel peptides combined with one of the other therapies, as shown in Figure 10, for effective cancer treatment.

## 4. Method and Materials

### 4.1. Cells, Reagents, Antibodies, and Kits

MDA-MB-468 cells were purchased from ATCC collections, MDA-MB-468 (ATCC^®^ HTB-132™). Cells were cultured in DMEM media with 10% foetal bovine serum, 2 mM L-glutamine purchased from Life Technologies Co., Paisley, UK, and supplemented with penicillin–streptomycin from Sigma-Aldrich Chemie GmbH, Taufkirchen, Germany. DAPI (4′,6-diamidino-2-phenylindole) obtained from Thermo-Fisher Scientific, Waltham, MA, USA was used in cell viability assays to stain the nucleic acid. Human anti-CD25-FITC (Cat. No.: 21810253) and anti-CD3-APC IgG1 (Cat. No.: 21810036) from ImmunoTools (Friesoythe, Germany) were used in T-cells activity assay. Anti-EGFR antibody [EP38Y] (ab52894) and goat anti-rabbit IgG H&L (FITC) (ab97050) were purchased from Abcam and used in tumour-killing assay. For the quantification of cytokines, a Custom ProcartaPlex Multiplex Panel manufactured by Thermo-Fisher Scientific was used. SEA protein was overexpressed and purified as described by Goda et al. using the gene with accession number KY594411, synthesised by Geneart GmbH (Regensburg, Germany) [13]. SEA-based peptides and their derivatives were synthesised and supplied by either GenScript (Piscataway Township, NJ, USA) or Proteogenix (Schiltigheim, France).

### 4.2. Design and Synthesis of Superantigen Peptides

Twenty superantigen-based peptides were designed to cover the full length of the SEA superantigen. Each peptide is 50 amino acids long and overlaps with the next peptide by 40 amino acids. GenScript (Piscataway Township, NJ, USA) chemically synthesised the designed peptides. The peptides were dissolved in water or DMSO based on GenScript’s recommendations.

### 4.3. Assessment of T-Cell Activation

Fresh blood from a healthy donor was used to extract peripheral blood mononuclear cells (PBMC). PBMCs were isolated using Lymphoprep solution as recommended by the manufacturing company (Stemcell Technologies, Cologne, Germany). In a 96-well plate, 2.5 × 10^5^ PBMC cells were seeded per well and treated with 30 µM of peptides, full superantigen, or controls. Treated cells were incubated for 72 h at 37 °C under 5% CO_2_ and humidified atmosphere. After incubation, the cells were collected and stained with fluorescently labelled anti-CD25 and anti-CD3 antibodies. T-cell activation was assessed by gating for CD3+CD25+ lymphocytes (Appendix A) using Accuri C6 flow cytometer (BD Biosciences, Workingham, UK). The percentage of CD3+CD25+ cells represents the percentage of activated T-cells. 

### 4.4. Design of Peptide P15 Derivatives

To study P15 further and to map the region responsible for the superantigenicity, we designed eight derivatives of P15. Four derivatives were designed by introducing amino acids from both the N and C terminals of P15, whereas the other four derivatives were designed by replacing aspartic acid (D) with alanine (A) to neutralise the acidic charge. The designed peptides were chemically synthesised by GenScript (Piscataway Township, NJ, USA). The peptides were dissolved in water as recommended by GenScript. T-cell activation analysis in response to 30 µM of P15 derivatives was carried out using the same method as described above.

### 4.5. Synthesis of Tumour-Targeted SEA-Based Peptides

SEA superantigen peptides P5, P14, and P15 were linked from the N’ or C’ terminal to TGFαL3 (VCHSGYVGARCEHADLL) with a flexible linker (GGSGSGGG). The final amino acid sequences were sent for chemical synthesis to ProteoGenix (Schiltigheim, France). Following ProteoGenix’s recommendations, acetonitrile was used to dissolve the peptide conjugates.

### 4.6. Tumour Cell Binding Assay

Tumour-targeted SEA-based peptides were evaluated for binding to EGFR receptors expressed on MDA-MB-468 by blocking anti-EGFR antibodies from binding to EGFR. In 96-well plates, 10^4^ cells per well were seeded and incubated at 37 °C, 5%CO_2_ overnight. The media was removed the following day, and 3.8% formaldehyde solution in PBS was added for 15 min at RT to fix the cells. The supernatant was discarded, and the cells were rinsed with PBST, followed by a blocking step using 3% BSA in PBS for 30 min at room temperature. Subsequently, the cells were washed with PBST and treated with 30 µM of fusion peptides or media control for 2 h at room temperature. After rinsing thrice with PBST, the cells were exposed to an anti-EGFR antibody from Abcam for 1 h at room temperature, followed by three additional PBST washes. The cells were stained with goat anti-rabbit IgG H&L -FITC and DAPI to visualise the nuclei for 30 min. For automated quantification of FITC- and DAPI-positive cells, ArrayScan™ XTI (Thermo Fisher Scientific, Waltham, MA, USA) was used. The percentage of peptide binding to tumour cells was determined by:$$\%\;Binding=\left(1-\frac{Response\;of\;Tumour\;cell\;treated\;with\;tumourtargeted\;SAg\;peptides}{Response\;of\;Tumour\;cell\;treated\;with\;antiEGFR}\right)\times 100$$

### 4.7. Antitumour Activity

MDA-MB-468 human breast cancer cell line was cultured at a density of 6.4 × 10^3^ cells per well in a 96-well plate and incubated at 37 °C in a 5% CO_2_ incubator overnight to allow cell adherence. Cells were then exposed to either 30 µM of SEA superantigen, SEA-derived peptides, cancer-targeting superantigen peptides, or solvent controls. After treatment, the cells were co-cultured with 3.2 × 10^4^ PBMCs/well, and the mixture was incubated for 48 h under the same conditions. Post-incubation, the cell lysates were collected for cytokine analysis. The cells were fixed using 3.8% formaldehyde solution for ten minutes and stained with DAPI. Viable cancer cells were quantified using the ArrayScan™ XTI instrument from Thermo Fisher Scientific (Waltham, MA, USA), which automates the measurement of DAPI-positive nuclei while excluding apoptotic nuclei based on characteristics such as reduced size, nuclear fragmentation and elevated chromatin intensities [41]. PBMCs were similarly excluded based on their size. Cell counts were expressed as a percentage of the total count from wells containing tumour cells and PBMCs.

### 4.8. Detection of Cytokines

Cell supernatant collected from the antitumour activity assay above was used to determine the production of various cytokines in response to peptide treatments, full SAgs, or control using a Custom ProcartaPlex Multiplex Panel ELISA kit, CellCarta Montreal, Canada. The assay was conducted according to the manual provided with the kit to measure levels of IL-1β, IL-2, IL-6, IFNγ, TNFα, and IL-10 cytokines with a FlexMap 3D instrument by Luminex, Austin, TX, USA.

### 4.9. MTT Assay

The toxicity of the fusion peptides was assessed on PBMCs using an MTT assay. In each well of a 96-well plate, 250,000 cells were seeded and exposed to fusion peptides, endotoxin-free SPEA, or solvent controls overnight at 37 °C with 5% CO_2_. Then, 20 µL of a 5 mg/mL MTT solution (obtained from Sigma-Aldrich, St. Louis, MO, USA) was added to each well, and the plates were incubated for another 4 h under the same conditions. The resulting formazan was solubilised with 200 μL of 20% SDS, and the absorbance at 570 nm was recorded. Cells treated with media were considered to represent 100% viability.

### 4.10. Statistics

The results are depicted as mean ± standard error of the mean (SEM) across “n” observations. Graphs were generated using GraphPad Prism 6 software, and statistical evaluations were performed using Student’s *t*-test. *p*-values < 0.05 indicate statistical significance.

## 5. Conclusions

We produced novel superantigen SEA-peptide agonists and SEA-peptide-agonist–EGF conjugates for the targeted treatment of EGFR-expressing cancers. We demonstrated that these peptides and their conjugates produced different cytokines and effectively killed cancer cells, while peptides and peptide conjugates did not affect healthy PBMCs. This work will open the door for further investigation such as conducting mutation studies on our isolated peptide agonists to produce more effective superantigen peptides for TTSP, as well as to fuse our peptides with other cancer-specific moieties to target different cancers. This study would pave the way for in vivo investigation of animals and humans.

## Figures and Tables

**Figure 1 ijms-25-09423-f001:**
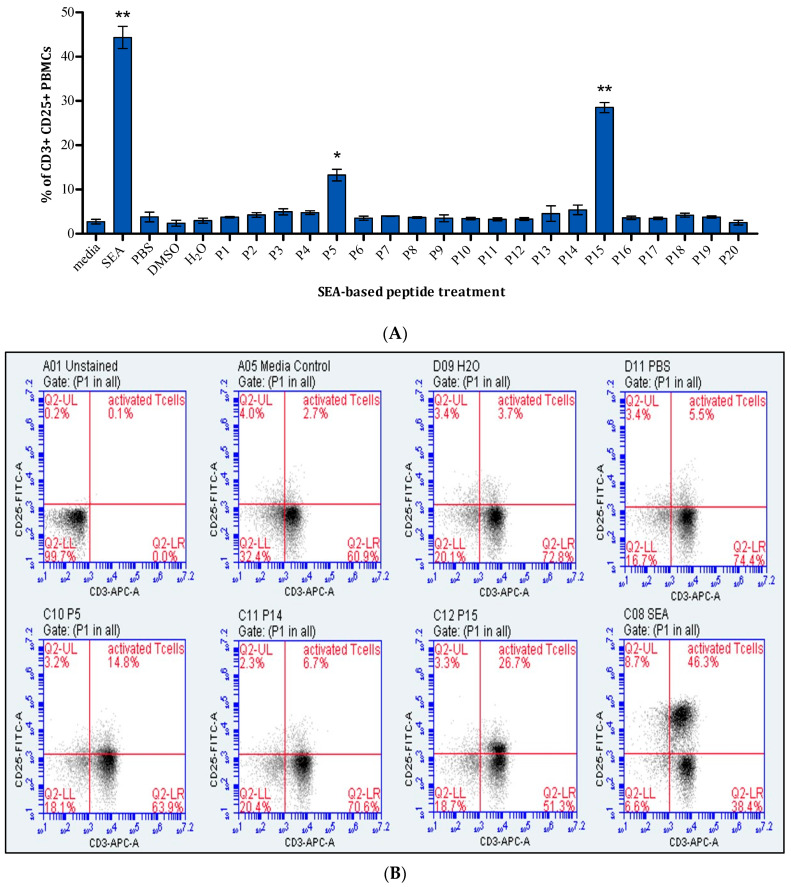
The effect of SEA-based peptides on T-cells. (**A**) Percentage of activated T-cells in response to 20 SEA-based peptides presented as mean ± standard error of the mean (SEM) from three independent experiments (n = 3). Statistical significance is determined using Student’s *t*-test and indicated as follows: * *p* < 0.05, ** *p* < 0.01, compared to the corresponding solvent control. (**B**) Representative dot plots of selected peptides response pre-gated for lymphocytes and stained with anti-CD3 and anti-CD25. It is worth mentioning that anti-CD4, anti-CD8, and appropriate MHC-II tetramers or dextramers should have been used to ensure that the antitumour activity is CD4+ TCR-dependent or bystander T-cell activation.

**Figure 2 ijms-25-09423-f002:**
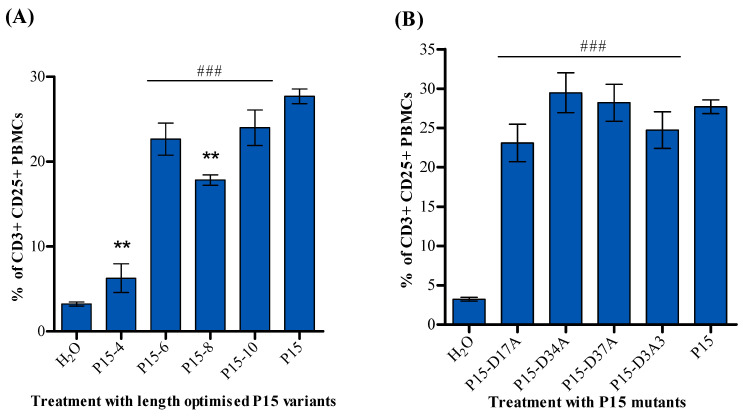
The effect of P15 peptide derivatives on T-cells. Percentage of activated T-cells resulting from PBMCs’ treatment with (**A**) length-optimised P15 variants and (**B**) P15 mutants, presented as mean ± standard error of the mean (SEM) of four independent experiments (n = 4). Statistical significance is determined using Student’s *t*-test and indicated as follows: ** *p* < 0.01, compared to peptide P15, whereas ### indicates *p* < 0.001 when compared to the solvent control. (**C**) Representative dot plots of selected peptides response pre-gated for lymphocytes and stained with anti-CD3 and anti-CD25.It is worth mentioning that anti-CD4, anti-CD8, and appropriate MHC-II tetramers or dextramers should have been used to ensure that the antitumour activity is CD4+ TCR-dependent or bystander T-cell activation.

**Figure 3 ijms-25-09423-f003:**
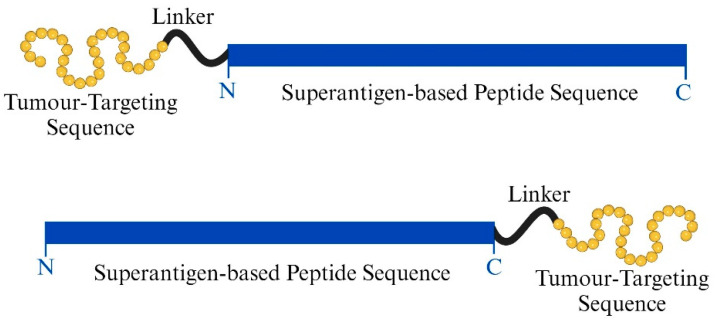
Design and synthesis of tumour-targeted SEA peptide construct.

**Figure 4 ijms-25-09423-f004:**
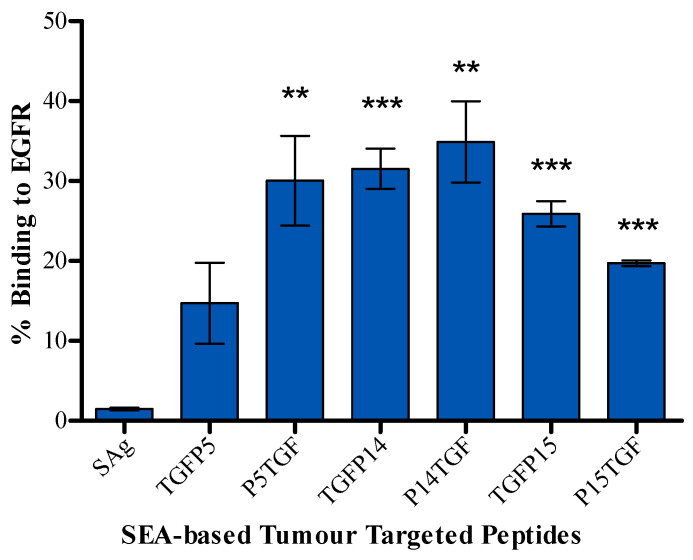
Binding of SEA-based tumour-targeted peptides to EGFR. Results are displayed as mean percentage ± standard error of the mean (SEM) from three experiments where statistical significance is determined using Student’s *t*-test and indicated as ** *p* < 0.01, *** *p* < 0.001 compared to the full superantigen.

**Figure 5 ijms-25-09423-f005:**
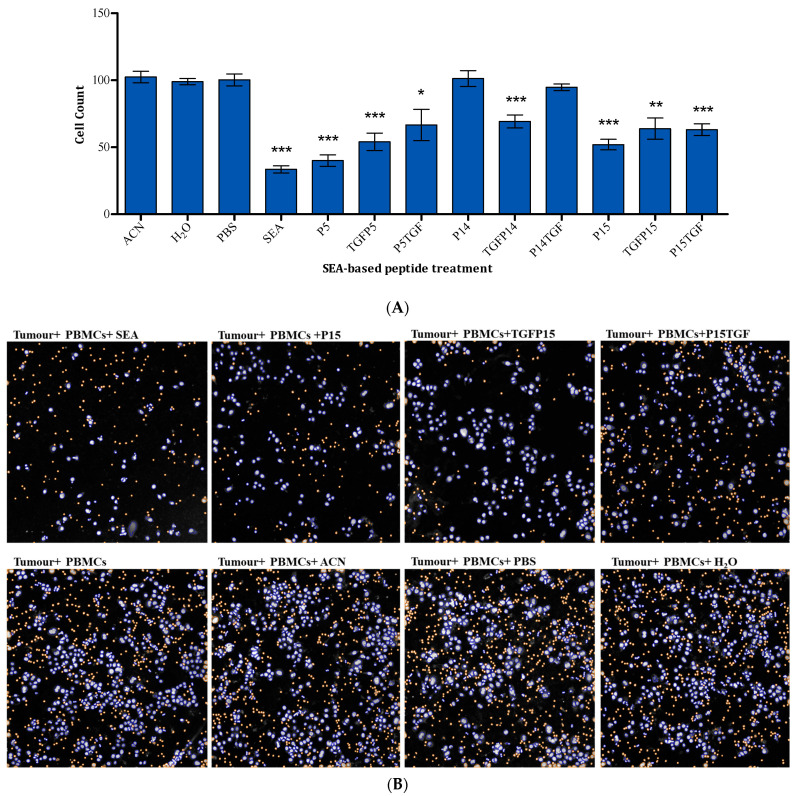
T-cell-dependent tumour-killing effect of SEA-based peptides and tumour-targeted superantigen peptides on MDA-MB-468 when co-cultured with PBMCs. (**A**) Mean of DAPI-positive nuclei as a percentage of TP ± standard error of the mean (SEM) of five experiments. Statistical significance is determined using Student’s *t*-test and indicated as: * *p* < 0.05, ** *p* < 0.01, *** *p* < 0.001, compared to the corresponding solvent. (**B**) Representative images of selected samples, acquired using ArrayScan™ XTI instrument, of DAPI-stained nuclei of viable cells. Blue represents DAPI-positive MDA-MB-468 cells, while orange indicates excluded PBMCs.

**Figure 6 ijms-25-09423-f006:**
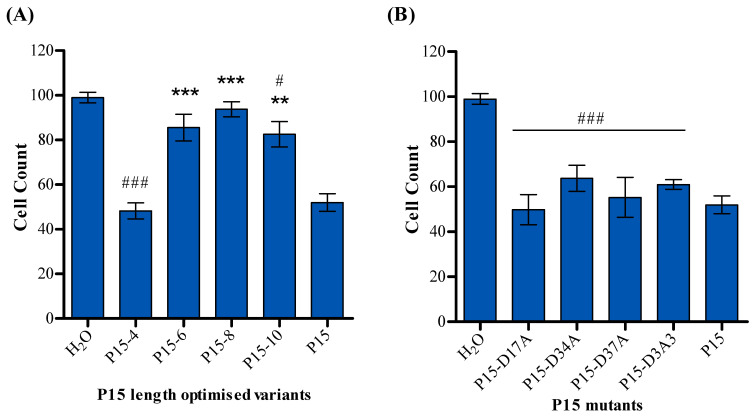
Effect of P15 peptide derivatives on MDA-MB-468 cell viability in a mixed culture with PBMCs. (**A**) P15 length-optimised variants, (**B**) P15 mutants. Results are presented as the mean of DAPI-positive nuclei calculated as a percentage of negative control samples containing tumour cells and PBMCs ± standard error of the mean (SEM) of six experiments. Statistical significance is determined using Student’s *t*-test and indicated as follows: ** *p* < 0.01 and *** *p* < 0.001, compared to peptide P15, whereas # indicates *p* < 0.05 and ### indicates *p* < 0.001 when compared to the solvent control.

**Figure 7 ijms-25-09423-f007:**
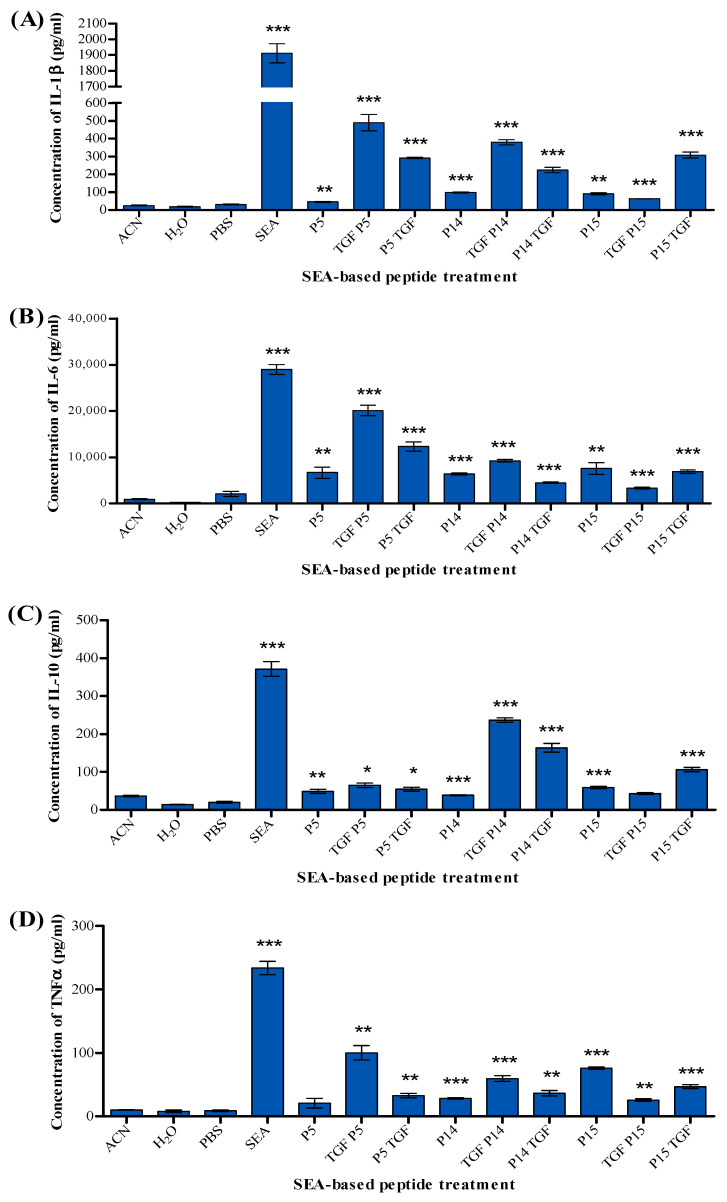
Level of cytokines produced in response to SEA-based peptides and tumour-targeted peptides in a co-culture of PBMCs and MDA-MB-468 cells. Panels show the concentration of (**A**) IL-1β, (**B**) IL-6, (**C**) IL-10, and (**D**) TNFα. Results are presented as mean concentrations in pg/mL ± standard error of the mean (SEM) from three experiments. Statistical significance is determined using Student’s *t*-test and indicated by * *p* < 0.05, ** *p* < 0.01, and *** *p* < 0.001, compared to treatment with ACN for tumour-targeted peptides and PBS for SEA.

**Figure 8 ijms-25-09423-f008:**
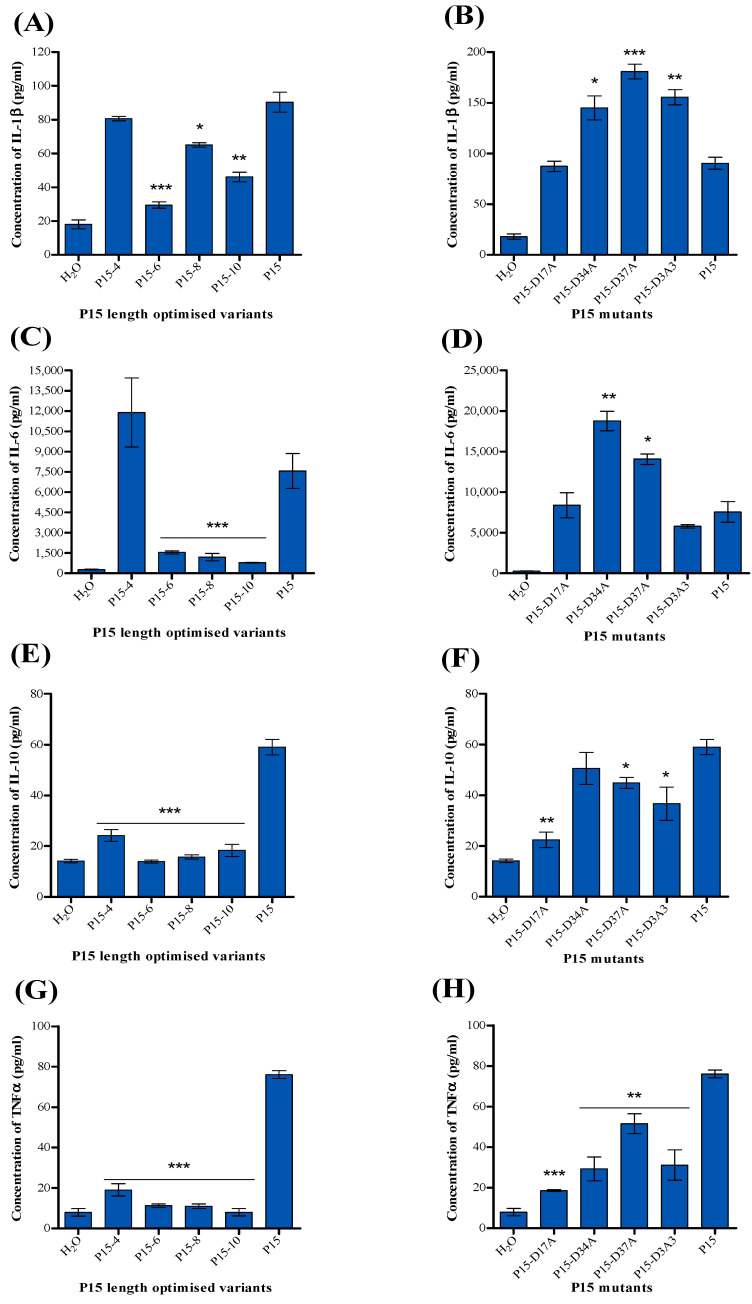
Level of cytokines produced after treatment with P15 derivatives, length-optimised and mutants, in a co-culture of PBMCs and MDA-MB-468 cells. Panels show the concentration of (**A**,**B**) IL-1β, (**C**,**D**) IL-6, (**E**,**F**) IL-10, and (**G**,**H**) TNFα. Results are presented as mean concentrations in pg/mL ± standard error of the mean (SEM) from three experiments. Statistical significance is determined using Student’s *t*-test and indicated by * *p* < 0.05, ** *p* < 0.01, and *** *p* < 0.001, compared to treatment with the P15 peptide.

**Figure 9 ijms-25-09423-f009:**
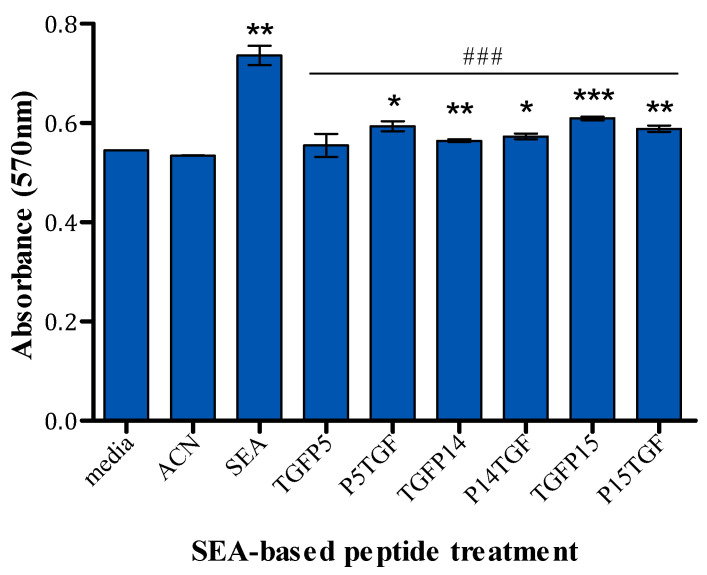
Impact of SEA-based tumour-targeting peptides on PBMC viability. The results are presented as mean absorbance ± standard error of the mean (SEM) where n = 4. Statistical significance is determined using Student’s *t*-test and indicated as * *p* < 0.05, ** *p* < 0.01, *** *p* < 0.001, compared to the media control. ### indicates *p* < 0.001 when compared to SEA superantigen.

**Figure 10 ijms-25-09423-f010:**
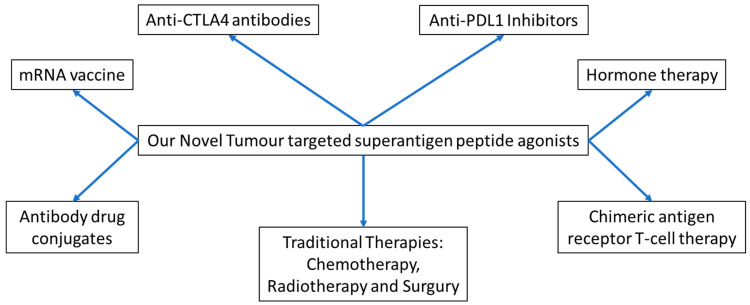
Possible use of our peptide therapy combined with other therapies.

**Table 1 ijms-25-09423-t001:** SEA-based peptide sequences and their solvents.

Label	Sequence	Solvent
P1	MSEKSEEINEKDLRKKSELQGTALGNLKQIYYYNEKAKTENKESHDQFLQ	H_2_O
P2	KDLRKKSELQGTALGNLKQIYYYNEKAKTENKESHDQFLQHTILFKGFFT	H_2_O
P3	GTALGNLKQIYYYNEKAKTENKESHDQFLQHTILFKGFFTDHSWYNDLLV	H_2_O
P4	YYYNEKAKTENKESHDQFLQHTILFKGFFTDHSWYNDLLVDFDSKDIVDK	H_2_O
P5	NKESHDQFLQHTILFKGFFTDHSWYNDLLVDFDSKDIVDKYKGKKVDLYG	H_2_O
P6	HTILFKGFFTDHSWYNDLLVDFDSKDIVDKYKGKKVDLYGAYYGYQCAGG	DMSO
P7	DHSWYNDLLVDFDSKDIVDKYKGKKVDLYGAYYGYQCAGGTPNKTACMYG	H_2_O
P8	DFDSKDIVDKYKGKKVDLYGAYYGYQCAGGTPNKTACMYGGVTLHDNNRL	H_2_O
P9	YKGKKVDLYGAYYGYQCAGGTPNKTACMYGGVTLHDNNRLTEEKKVPINL	H_2_O
P10	AYYGYQCAGGTPNKTACMYGGVTLHDNNRLTEEKKVPINLWLDGKQNTVP	H_2_O
P11	TPNKTACMYGGVTLHDNNRLTEEKKVPINLWLDGKQNTVPLETVKTNKKN	H_2_O
P12	GVTLHDNNRLTEEKKVPINLWLDGKQNTVPLETVKTNKKNVTVQELDLQA	H_2_O
P13	TEEKKVPINLWLDGKQNTVPLETVKTNKKNVTVQELDLQARRYLQEKYNL	H_2_O
P14	WLDGKQNTVPLETVKTNKKNVTVQELDLQARRYLQEKYNLYNSDVFDGKV	H_2_O
P15	LETVKTNKKNVTVQELDLQARRYLQEKYNLYNSDVFDGKVQRGLIVFHTS	H_2_O
P16	VTVQELDLQARRYLQEKYNLYNSDVFDGKVQRGLIVFHTSTEPSVNYDLF	DMSO
P17	RRYLQEKYNLYNSDVFDGKVQRGLIVFHTSTEPSVNYDLFGAQGQYSNTL	H_2_O
P18	YNSDVFDGKVQRGLIVFHTSTEPSVNYDLFGAQGQYSNTLLRIYRDNKTI	H_2_O
P19	QRGLIVFHTSTEPSVNYDLFGAQGQYSNTLLRIYRDNKTINSENMHIDIY	DMSO
P20	TEPSVNYDLFGAQGQYSNTLLRIYRDNKTINSENMHIDIYLYTS	DMSO

**Table 2 ijms-25-09423-t002:** Amino acid sequence of P15 derivatives.

Label	Sequence
P15-4	KTNKKNVTVQELDLQARRYLQEKYNLYNSDVFDGKVQRGLIV
P15-6	NKKNVTVQELDLQARRYLQEKYNLYNSDVFDGKVQRGL
P15-8	KNVTVQELDLQARRYLQEKYNLYNSDVFDGKVQR
P15-10	VTVQELDLQARRYLQEKYNLYNSDVFDGKV
P15-D17A	LETVKTNKKNVTVQEL**A**LQARRYLQEKYNLYNSDVFDGKVQRGLIVFHTS
P15-D34A	LETVKTNKKNVTVQELDLQARRYLQEKYNLYNS**A**VFDGKVQRGLIVFHTS
P15-D37A	LETVKTNKKNVTVQELDLQARRYLQEKYNLYNSDVF**A**GKVQRGLIVFHTS
P15-D3A3	LETVKTNKKNVTVQEL**A**LQARRYLQEKYNLYNS**A**VF**A**GKVQRGLIVFHTS

## Data Availability

All data are available with the authors and will be provided if requested.

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
