# Peer review of "Generation of Novel Tumour-Selective SEA Superantigen-Based Peptides with Improved Safety and Efficacy for Precision Cancer Immunotherapy"

_ijms, 2024, doi:10.3390/ijms25179423_

Round 1

Reviewer 1 Report

Comments and Suggestions for Authors

This article presents an approach to harness T cell immunity using superantigen-based peptides derived from Staphylococcal enteroxin type A (SEA), and later as an anti-cancer peptide-TGFαL3-SEA conjugate. While addressing the need for a safer alternative by optimising the peptide length and mutating residues of lead candidate P15 (141-190).

Strength

The authors have identified SEA-based peptides with antigenic activities and as peptide-TGFαL3 conjugates to become cancer killing by redirecting the T cells to the cancer cells as T cell engager.

Weaknesses

This work is highly similar to a recently published work by the same group: https://www.ncbi.nlm.nih.gov/pmc/articles/PMC10341475/

The authors designed a fusion polypeptide (TGFαL3-SEA) to redirect activated T cells to the MDA-MB-468 triple negative breast cancers. This is a similar approach to the article "Evaluation of Antitumor Activity of TGFαL3-SEB as a Ligand-Targeted Superantigen. Technol Cancer Res Treat. 2016 Apr;15(2):215-26." 

Figures 1 and 2: The flow cytometry data primarily focuses on SEA-derived peptides with insufficient panel markers to properly characterise CD4+ T subsets in the PBMCs. Without anti-CD4, anti-CD8 and appropriate MHC-II tetramers or dextramers, it is unclear if the anti-tumour activity is CD4+ TCR-dependent or bystander T cell activation.

P14 is not a potent peptide in Figures 1 and 5. The sentence " We also found that P14, which overlaps with (potent) P15 managed to produce the same cytokines" is contradicting (pp7).

Figure 7: The molecular orientation of peptide:TGF or TGF:peptide can result in differing levels of cytokines. This suggests antigen-processing that may be unique to the HLA haplotypes of the PBMCs. For example, Figure 7D: TNFα in P14 is less than P15 but TNFα in TGFP14 is more than TGFP15. 

The HLA of the PBMCs should be typed to ascertain that TNKKMVTAQELD on P15 is likely antigenic combined with NetMHCII prediction.

The use of TCGαL3 sequence to target the TGFalpha binding site of EGFR is neither clearly stated nor cited. 

Comments on the Quality of English Language

No major issues except for minor spacing issues and some spelling mistakes e.g. Supplimentary 

Author Response

Reviewer 1:

This article presents an approach to harness T cell immunity using superantigen-based peptides derived from Staphylococcal enteroxin type A (SEA), and later as an anti-cancer peptide-TGFαL3-SEA conjugate. While addressing the need for a safer alternative by optimising the peptide length and mutating residues of lead candidate P15 (141-190).

We thank you for your time and attention in reviewing our paper and for your valuable comments, which will improve our manuscript.  

Strength

The authors have identified SEA-based peptides with antigenic activities and as peptide-TGFαL3 conjugates to become cancer killing by redirecting the T cells to the cancer cells as T cell engager.

We thank you for your kind comment.

Weaknesses

  • This work is highly similar to a recently published work by the same group: https://www.ncbi.nlm.nih.gov/pmc/articles/PMC10341475

The SPEA (used in our previous study) and the SEA, in this current investigation, share low amino acid homology. The production of peptides from each superantigen with T cell activation and the capability of killing cancer cells could be used consecutively in cancer treatment to minimise immunogenicity and drug resistance. In addition, we carried out new work in the current manuscript, such as peptide modifications and peptide mutagenesis for peptide optimisation.  An explanation was added to the discussion.

  • The authors designed a fusion polypeptide (TGFαL3-SEA) to redirect activated T cells to the MDA-MB-468 triple negative breast cancers. This is a similar approach to the article "Evaluation of Antitumor Activity of TGFαL3-SEB as a Ligand-Targeted Superantigen. Technol Cancer Res Treat. 2016 Apr;15(2):215-26." 

The work conducted in the referred paper was focused on full-length SEB superantigen fused with TGFαL3. It is well documented that the use of full superantigen in cancer treatment has severe side effects due to its toxicity. Our work focused on producing superantigen peptide agonists with minimum lethality and modified peptide agonists to overcome the pitfalls of using full superantigens. Also, our work opened the door to linking different cancer-specific sequences to target our peptides to any cancer cells.

  • Figures 1 and 2: The flow cytometry data primarily focuses on SEA-derived peptides with insufficient panel markers to properly characterise CD4+ T subsets in the PBMCs. Without anti-CD4, anti-CD8 and appropriate MHC-II tetramers or dextramers, it is unclear if the anti-tumour activity is CD4+ TCR-dependent or bystander T cell activation.

Thank you for raising this point. We have added this comment in the legend of both figures.

P14 is not a potent peptide in Figures 1 and 5. The sentence " We also found that P14, which overlaps with (potent) P15 managed to produce the same cytokines" is contradicting (pp7).

Thank you. The sentence confusing was deleted.

  • Figure 7: The molecular orientation of peptide:TGF or TGF:peptide can result in differing levels of cytokines. This suggests antigen-processing that may be unique to the HLA haplotypes of the PBMCs. For example, Figure 7D: TNFα in P14 is less than P15 but TNFα in TGFP14 is more than TGFP15. 

Thank you for clarification. 

  • The HLA of the PBMCs should be typed to ascertain that TNKKMVTAQELD on P15 is likely antigenic combined with NetMHCII prediction.

The sequence TNKKMVTAQELD is a conservative region in most superantigens and is already known to have bind with CD28 and is responsible for massive cytokine release

https://pubmed.ncbi.nlm.nih.gov/21931534/

https://pubmed.ncbi.nlm.nih.gov/10742148/  

  • The use of TCGαL3 sequence to target the TGFalpha binding site of EGFR is neither clearly stated nor cited.

Please refer to the paragraph, which includes references 32-40. Thank you

  • Comments on the Quality of English Language: No major issues except for minor spacing issues and some spelling mistakes e.g. Supplimentary

Thank you. In addition, we carried out a complete revision for confirmation.

Reviewer 2 Report

Comments and Suggestions for Authors

In the present manuscript, the authors explore the option of extracting the best properties of SEA-superantigen for therapeutic application, by expressing a number of pertaining peptides and fusing them with a targeting moiety. They discovered 2 peptides that indeed have shown certain activity in stimulating PBMCs, which was attenuated in comparison with complete SEA.  They then investigate the response to these molecules, in part fused with specific targeting sequences,  in an EGFR-positive cell line, including killing assays with PBMCs. The main aim of such molecules is to decrease the generalized unspecific cytokine response, which they have also tested here.

The studies in this direction are very valuable because they open an avenue for a novel class of therapeutics, by employing cancer cell targeting and fine-tuning specific (at least in part)  T-cell activation. However, there are several points that should be addressed:

-          The manuscript is very similar to the one describing the activity of SPEA-derived peptides from the same authors. The present manuscript would profit from the information, why certain mutations were constructed (a rationale to mutant design), and (at least a tentative) explanation of data in connection with their activity, with consideration of structural properties of the superantigen. Comparison of findings in both cases would also be enriching, especially in the discussion section.

-          Site-mutations: do they add value to the knowledge conveyed in this manuscript? (maybe this point is clear when the reason for their design is explained, as asked for in the previous point)

-          Methodology: data on the activity of the compounds for an antigen-positive cell line is presented, but it is important to have the read-out of the effect on the antigen-negative cell line in the same assays.

-          Structure of the manuscript: important background information first appears in the discussion section. This should be a part of Introduction (please also see the specific remarks below).

-          Statistical analyses: which method was used for which assay? This should also be mentioned in the individual Figure legends.

-          Language: spelling is not perfect and some sentences not grammatically correct. The manuscript should be carefully re-read by all authors and especially by the senior authors

-          References require correction: 14 and 43 are the same

Please find below a list of remarks which I hope you will find helpful.

Author’s emails: “ [email protected]

Page 1: “TGFαL3“ – please explain the abbreviation and the context

Page 2:” 2mM of L-Glutamine 200 mM (100x)” – please correct to 2 mM L-glutamine

Page 2: SPEA-based peptides: the abbreviation should be explained

Page 2: all antibodies should be cited with RRIDs, if possible, or otherwise with source and catalogue number

Page 2: “were from abcam were used in tumor killing assay.” – please correct the sentence

Table 1: it is not explained why a certain sequence is highlighted in yellow; the information is given later in the text, but should be rendered at this point

Page 3: “for 72h at 37°C 5% CO2” – under 5% CO (2 In subscript), and probably this was in humidified atmosphere

Page 3: “we designed eight derivatives to P15” – of P15

Page 4: “Whereas, the other four derivatives were designed by replacing Aspartic acid (D) to Alanine (A).“ – how were the mutations chosen?

Page 4: “0.2 mg/ml of fusion peptides” – please cite all concentrations in molar values

Page 4: “was carried using the same method above” – was carried out using the same method as described above

Page 5: “H2O“ – 2 in subscript

Page 7: “…respectively. While TGFP15 and P15TGF had slightly lower yet significant binding effect“ – please make to one sentence

Page 9, Figure 5: The killing of antigen-negative cell line should be investigated.

Page 13 and 14: “Contrary to the conventional antigen…” and ending with: page 14, “reliability and validity of our findings.” – should be a part of the introduction

Page 14: “could activate and proliferate human T-cells and produce several cytokines,” – surely, the peptides cause proliferation and cytokine production?

Page 14: “the release of massive cytokines” – massive release of cytokines

Page 14:” behave as noted by the full SEA superantigen.” – in which way and which assays show this?

Page 14: “have a similar profile of the full superantigens” – in which way are they similar to SEA?

Page 14: “It was shown that the CD28 homodimer interface” – to “for future work on peptide bioavailability as a drug.” – should be a part of the introduction

Page 14:” significant decrease in the percentage of activated T-cells” – a significantly lower percentage, there was not a decrease

Page 15:” a safer for of a superantigen” – a safer form?

Page 15: from “In this work, we employ…” to “with the addition of a flexible linker „ – should be part of the introduction

Page 15:” from “Figure 10 shows many of the known strategies for treating different types of cancer “ – it is not specified how and why the findings can support combination therapies, the strategies are simply listed and do not add any value to the manuscript.

Page 16: “Also, to fuse our peptides with other cancer-specific moieties to target treating different cancers. This study would pave the way for in vivo investigation of animals and humans “ – due to wrong grammatical structure the message is lost. 

Author Response

Reviewer 2:

Comments and Suggestions for Authors

In the present manuscript, the authors explore the option of extracting the best properties of SEA-superantigen for therapeutic application, by expressing a number of pertaining peptides and fusing them with a targeting moiety. They discovered 2 peptides that indeed have shown certain activity in stimulating PBMCs, which was attenuated in comparison with complete SEA.  They then investigate the response to these molecules, in part fused with specific targeting sequences,  in an EGFR-positive cell line, including killing assays with PBMCs. The main aim of such molecules is to decrease the generalized unspecific cytokine response, which they have also tested here.

The studies in this direction are very valuable because they open an avenue for a novel class of therapeutics, by employing cancer cell targeting and fine-tuning specific (at least in part)  T-cell activation. However, there are several points that should be addressed:

We would like to thank the reviewer for reviewing our manuscript and for suggesting very valuable comments that will enhance it. Please find below the Authors’ response to your comments in red.

-          The manuscript is very similar to the one describing the activity of SPEA-derived peptides from the same authors. The present manuscript would profit from the information, why certain mutations were constructed (a rationale to mutant design), and (at least a tentative) explanation of data in connection with their activity, with consideration of structural properties of the superantigen. Comparison of findings in both cases would also be enriching, especially in the discussion section.

The SPEA (used in our previous study) and the SEA, in this current investigation, share low amino acid homology. The production of peptides from each superantigen with T cell activation and the capability of killing cancer cells could be used consecutively in cancer treatment to minimize immunogenicity and drug resistance. In addition, we carried out new work in the current manuscript, such as peptide modifications and peptide mutagenesis for peptide optimization. 

This has been added to the discussion; thank you

-          Site-mutations: do they add value to the knowledge conveyed in this manuscript? (maybe this point is clear when the reason for their design is explained, as asked for in the previous point)

The mutation was chosen randomly but based on changing acidic or basic amino acid to neutral amino acid to study the effect. This point will be discussed in the discussion; thank you.

-          Methodology: data on the activity of the compounds for an antigen-positive cell line is presented, but it is important to have the read-out of the effect on the antigen-negative cell line in the same assays.

Thank you for raising this point.

The activity of our peptides was tested through a T-cell activation assay and an anti-tumour assay. The T-cell activation assay was carried out on PBMCs (antigen-negative cells), while the tumour-killing assay was conducted on a co-culture of PBMCs and tumour cells.

We agree with the reviewer that adding a healthy cell line would have added more value to our results. We, however, carried out the MTT assay to test the safety of our peptide conjugates on PBMCs and showed that the peptides significantly reduced toxicity on PBMCs.

Our work will open the door for future studies and investigation of our novel peptides.

-          Structure of the manuscript: important background information first appears in the discussion section. This should be a part of Introduction (please also see the specific remarks below).

Modification has been carried out in the manuscript based on the reviewer's suggestion; thank you.

-          Statistical analyses: which method was used for which assay? This should also be mentioned in the individual Figure legends.

Modification has been carried out in the manuscript, accordingly.

-          Language: spelling is not perfect and some sentences not grammatically correct. The manuscript should be carefully re-read by all authors and especially by the senior authors

A complete revision of the English was carried out, and many corrections have been made.

-          References require correction: 14 and 43 are the same

References were corrected.

Please find below a list of remarks which I hope you will find helpful.

Author’s emails: “ [email protected]

It is correct in the word format but wrong in the PDF. Will ensure it is correct in the PDF as well. Thank you

Page 1: “TGFαL3“ – please explain the abbreviation and the context

The third loop (L3) of transforming growth factor-α (TGF-α). More explanation was mentioned in the paragraphs, including references 32-40.

Page 2:” 2mM of L-Glutamine 200 mM (100x)” – please correct to 2 mM L-glutamine

Modification has been carried out in the manuscript.

Page 2: SPEA-based peptides: the abbreviation should be explained

Done, thank you

Page 2: all antibodies should be cited with RRIDs, if possible, or otherwise with source and catalogue number

Modification has been carried out in the manuscript; thank you.

Page 2: “were from abcam were used in tumor killing assay.” – please correct the sentence

Modification has been carried out in the manuscript.

Table 1: it is not explained why a certain sequence is highlighted in yellow; the information is given later in the text, but should be rendered at this point

Explanation was added to the table ligand.

Page 3: “for 72h at 37°C 5% CO2” – under 5% CO (2 In subscript), and probably this was in humidified atmosphere

Modification has been carried out in the manuscript.

Page 3: “we designed eight derivatives to P15” – of P15

Modification has been carried out in the manuscript.

Page 4: “Whereas, the other four derivatives were designed by replacing Aspartic acid (D) to Alanine (A).“ – how were the mutations chosen?

As explained above, the mutation was chosen randomly, but acidic amino acid was replaced by neutral amino acid

Page 4: “0.2 mg/ml of fusion peptides” – please cite all concentrations in molar values

Modification has been carried out in the manuscript; thank you.

Page 4: “was carried using the same method above” – was carried out using the same method as described above

Modification has been carried out in the manuscript.

Page 5: “H2O“ – 2 in subscript

Modification has been carried out in the manuscript.

Page 7: “…respectively. While TGFP15 and P15TGF had slightly lower yet significant binding effect“ – please make to one sentence

Modification has been carried out in the manuscript.

Page 9, Figure 5: The killing of antigen-negative cell line should be investigated.

As we mentioned above, the activity of our peptides was tested through a T-cell activation assay and an anti-tumour assay. The T-cell activation assay was carried out on PBMCs (antigen-negative cells), while the tumour-killing assay was conducted on a co-culture of PBMCs and tumour cells.

We agree with the reviewer that adding a healthy cell line would have added more value to our results. We, however, carried out the MTT assay to test the safety of our peptide conjugates on PBMCs and showed that the peptides significantly reduced toxicity on PBMCs.

Our work will open the door for future studies and investigation of our novel peptides.

Page 13 and 14: “Contrary to the conventional antigen…” and ending with: page 14, “reliability and validity of our findings.” – should be a part of the introduction

Done, thank you

Page 14: “could activate and proliferate human T-cells and produce several cytokines,” – surely, the peptides cause proliferation and cytokine production?

Done, thank you

Page 14: “the release of massive cytokines” – massive release of cytokines

Modification has been carried out in the manuscript.

Page 14:” behave as noted by the full SEA superantigen.” – in which way and which assays show this?

Clarified in the text, thank you

Page 14: “have a similar profile of the full superantigens” – in which way are they similar to SEA?

Clarified in the text, thank you

Page 14: “It was shown that the CD28 homodimer interface” – to “for future work on peptide bioavailability as a drug.” – should be a part of the introduction

I would kindly request to keep this part in the discussion because of the following sentences after this part.

Page 14:” significant decrease in the percentage of activated T-cells” – a significantly lower percentage, there was not a decrease

Modification has been carried out in the manuscript.

Page 15:” a safer for of a superantigen” – a safer form?

Modification has been carried out in the manuscript.

Page 15: from “In this work, we employ…” to “with the addition of a flexible linker „ – should be part of the introduction

Done, accordingly, thank you

Page 15:” from “Figure 10 shows many of the known strategies for treating different types of cancer “ – it is not specified how and why the findings can support combination therapies, the strategies are simply listed and do not add any value to the manuscript.

More explanation was added to clarify this part; thank you

Page 16: “Also, to fuse our peptides with other cancer-specific moieties to target treating different cancers. This study would pave the way for in vivo investigation of animals and humans “ – due to wrong grammatical structure the message is lost. 

A clarification was carried out in the manuscript; thank you

Round 2

Reviewer 1 Report

Comments and Suggestions for Authors

There is still inconsistency in the data presented between Fig 2, Fig 5 and Fig 6. A multiplexed AIM assay should have been performed to delineate either activated CD4 or CD8 T cell effects. It is still unclear why P15-4 which has a decreased percentage of activated T cells can still result in a decreased % MDA-MB-468 cell viability comparable to P15 (Fig 6A). Similarly, despite designing P15 Ala mutants, the P15 mutations did yield significant biological outcomes compared to P15, not solvent control (Fig 6B). 

Comments on the Quality of English Language

Minor edit for page 14 "Activated T cells in I'm k".

Author Response

                                                           Reviewer 1 round 2 response.

There is still inconsistency in the data presented between Fig 2, Fig 5 and Fig 6. A multiplexed AIM assay should have been performed to delineate either activated CD4 or CD8 T cell effects. It is still unclear why P15-4 which has a decreased percentage of activated T cells can still result in a decreased % MDA-MB-468 cell viability comparable to P15 (Fig 6A).

The above points are valid. However, the peptide's structure, sequence, binding, and folding determine its activity. Also, superantigens or superantigen agonists, after binding to T cells, induce the production of a broad range of cytokines, including tumour necrosis factor (TNF), interleukin 2 (IL-2), interleukin 1 (IL-1), IL-4, IL-6, and gamma interferon. Although P15-4 produces less T-cell activation, it might stimulate the production of cytokines in a slightly different ratio than P15, which could be an avenue for more effective cancer cell killing. This point was also added to the discussion on pages 26-27.

Similarly, despite designing P15 Ala mutants, the P15 mutations did yield significant biological outcomes compared to P15, not solvent control (Fig 6B). 

We have acknowledged this point in the manuscript in the last paragraph on page 25. 

Minor edit for page 14 "Activated T cells in I'm k".

Corrected, thank you.

Another point

Regarding the point the editor raised in the email that reviewer 1 felt that the article was not sufficiently novel, we mentioned the following in our discussion to respond to this point and to highlight the paramount importance of carrying out the current work:

“The SPEA and SEA superantigens share low amino acid homology. The peptides produced in our previous work on SPEA and the current study on SEA with T-cell activation and the capability of killing cancer cells could be used consecutively in cancer treatment to minimise immunogenicity and drug resistance. In addition, we carried out new work in the current manuscript, such as peptide modifications and peptide mutagenesis for peptide optimisation”. 

Reviewer 2 Report

Comments and Suggestions for Authors

The authors have restructured the manuscript, improved the language for clarity, and placed their experimental achievements into a broader context. Please find below a list of only minor remarks which I hope you will find helpful.

Page 1: “ Low immunogenicity and immune suppression could explain this, which leads to limited tumour T-cell activation” – This can be explained by limited tumour T-cell activation resulting from low immunogenicity and immune suppression (word order)

Page 4: „Four derivatives were designed by introducing amino acids from both the N and C terminals of P15. Whereas the other four derivatives were designed by replacing Aspartic acid (D) with Alanine (A)” - Four derivatives were designed by introducing amino acids from both the N and C terminals of P15, whereas the other four derivatives were designed by replacing Aspartic acid (D) with Alanine (A) to neutralize the acidic charge.

Page 3: “DAPI (40’,6-diamidino-2-phenylindole)” – DAPI (4’,6-diamidino-2-phenylindole)

Page 3: Thermo Fisher Scientific

Page 4:” response to 30 uM of P15” – micromolar, please correct throughout the manuscript

Page 5: “Cell counts were expressed as a percentage of the total count from wells containing tumor cells and PBMCs (TP)” and Page 10: a percentage of untreated (TP) sample: why is this abbreviation necessary and it does not appear in the text or Figures otherwise.

Page 9: “using Student’s t-test andindicated” - and indicated

Page 9: that treatment with P15-6, P15-8 and P15-10 resulted in increased cell viability

Page 13: “as well as P15TGF had a significant increase in the absorbance compared to the solvent control which indicates cell proliferation. However, when compared to the full superantigen, these peptide conjugates showed a reduced cell proliferation effect; therefore, they were less toxic on PBMCs.” – P15TGF caused a significant increase in the absorbance… / peptide conjugates caused a reduced negative cell proliferation effect

Page 14:” causing G0–G1 cell” – line break

Author Response

Round 2 responses.

The authors have restructured the manuscript, improved the language for clarity, and placed their experimental achievements into a broader context. Please find below a list of only minor remarks which I hope you will find helpful.

We thank you for your detailed review of our manuscript and valuable corrections and suggestions.

Page 1: “ Low immunogenicity and immune suppression could explain this, which leads to limited tumour T-cell activation” – This can be explained by limited tumour T-cell activation resulting from low immunogenicity and immune suppression (word order)

Corrections were carried out, thank you.

Page 4: „Four derivatives were designed by introducing amino acids from both the N and C terminals of P15. Whereas the other four derivatives were designed by replacing Aspartic acid (D) with Alanine (A)” - Four derivatives were designed by introducing amino acids from both the N and C terminals of P15, whereas the other four derivatives were designed by replacing Aspartic acid (D) with Alanine (A) to neutralize the acidic charge.

Corrected, thank you.

Page 3: “DAPI (40’,6-diamidino-2-phenylindole)” – DAPI (4’,6-diamidino-2-phenylindole)

Corrected, thank you

Page 3: Thermo Fisher Scientific

Corrected, thank you

Page 4:” response to 30 uM of P15” – micromolar, please correct throughout the manuscript

All uM were replaced by µM; thank you.

Page 5: “Cell counts were expressed as a percentage of the total count from wells containing tumor cells and PBMCs (TP)” and Page 10: a percentage of untreated (TP) sample: why is this abbreviation necessary and it does not appear in the text or Figures otherwise.

Corrected, thank you.

Page 9: “using Student’s t-test andindicated” - and indicated

Corrected, thank you

Page 9: that treatment with P15-6, P15-8 and P15-10 resulted in increased cell viability

Modified, thank you

Page 13: “as well as P15TGF had a significant increase in the absorbance compared to the solvent control which indicates cell proliferation. However, when compared to the full superantigen, these peptide conjugates showed a reduced cell proliferation effect; therefore, they were less toxic on PBMCs.” – P15TGF caused a significant increase in the absorbance… / peptide conjugates caused a reduced negative cell proliferation effect

Corrected, thank you.

Page 14:” causing G0–G1 cell” – line break

Corrected, thank you.